# Unmanned Aerial Vehicle (UAV)-Based Remote Sensing for Early-Stage Detection of Ganoderma

**Parisa Ahmadi** [1,*] **, Shattri Mansor** [2] **, Babak Farjad** [3] **and Ebrahim Ghaderpour** [3]

1   Institute of Ocean and Earth Science (IOES), University of Malaya,
    Kuala Lumpur 50603, Wilayah Persekutuan Kuala Lumpur, Malaysia
2   Department of Civil Engineering, Faculty of Engineering, Universiti Putra Malaysia,
    Seri Kembangan 43400, Selangor, Malaysia; shattri@upm.edu.my
3   Department of Geomatics Engineering, Schulich School of Engineering, University of Calgary,
    2500 University Drive NW, Calgary, AB T2N 1N4, Canada; bfarjad@ucalgary.ca (B.F.);
    ebrahim.ghaderpour@ucalgary.ca (E.G.)
*   Correspondence: parisa@um.edu.my; Tel.: +1-(236)-869-0678

**Abstract:** Early detection of Basal Stem Rot (BSR) disease in oil palms is an important plantation management activity in Southeast Asia. Practical approaches for the best strategic approach toward the treatment of this disease that originated from Ganoderma Boninense require information about the status of infection. In spite of the availability of conventional methods to detect this disease, they are difficult to be used in plantation areas that are commonly large in terms of planting hectarage; therefore, there is an interest for a quick and delicate technique to facilitate the detection and monitoring of Ganoderma in its early stage. The main goal of this paper is to evaluate the use of remote sensing technique for the rapid detection of Ganoderma-infected oil palms using Unmanned Aerial Vehicle (UAV) imagery integrated with an Artificial Neural Network (ANN) model. Principally, we sought for the most representative mean and standard deviation values from green, red, and near-infrared bands, as well as the best palm circle radius, threshold limit, and the number of hidden neurons for different Ganoderma severity levels. With the obtained modified infrared UAV images at 0.026 m spatial resolution, early BSR infected oil palms were most satisfactorily detected with mean and standard deviation derived from a circle radius of 35 pixels of band green and near-infrared, 1/8 threshold limit, and ANN network by 219 hidden neurons, where the total classification accuracies achieved for training and testing the dataset were 97.52% and 72.73%, respectively. The results from this study signified the utilization of an affordable digital camera and UAV platforms in oil palm plantation, predominantly in disease management. The UAV images integrated with the Levenberg–Marquardt training algorithm illustrated its great potential as an aerial surveillance tool to detect early Ganoderma-infected oil palms in vast plantation areas in a rapid and inexpensive manner.

**Keywords:** ANN; Basal Stem Rot; remote sensor; Levenberg–Marquardt; UAV

## 1. Introduction

The palm oil industry is now the fourth largest contributor to the Malaysian economy [1] and plays an important role in other countries, such as Indonesia [2], Thailand [3], and Africa [4]. It is well documented that oil palm trees need a comprehensive and ongoing understanding of their current state, since oil palms are subjected to major pathogens that can threaten the production of palm oil. Among the crucial diseases that requires attention, Basal Stem Rot (BSR) is a root disease that has been estimated to cost as much as USD 500 million a year to oil palm producers in some Southeast Asian countries [5]. Hence, in order to improve planning and management decisions, information on BSR infected oil palms is needed. The identification of the disease, however, is very difficult due to the fact that there is no symptom at the early stages of infection. It is only when the disease

has reached a critical stage that symptoms of infection usually manifest, which include the following: rot of the base of the stem from where basidiocarps of Ganoderma Boninense rise, as well as the rot of the roots and chlorosis. At the advanced level, wilting and skirting of the older fronds could be observed [6], along with leveling of the crown and unopened spears and cracking of the stem [7].

Currently, the early identification of infected palms can be performed by some techniques, such as (I) the colorimetric strategy utilizing Ethylenediamine Tetra-acetic Acid (EDTA) [8], (II) Ganoderma-semi-selective media on agar plates [9], (III) Ganoderma-semi-selective media [10], (IV) Polymerase Chain Response (PCR) [11], (V) Ganoderma Selective Media (GSM) testing for any tainted tissues [12], and Volatile Organic Compounds (VOCs) [10]. Nevertheless, the majority of these techniques are tedious, costly, and not feasible for applications in plantation areas. A perfect strategy for the detection of diseases requires minimal sample preparation, preciseness, and the quality of being non-destructive [13].

In contrast to other remote sensing applications, a UAV-based method can be applied as a practical substitute due to lower cost funds, widespread utilization, better resolution (from many meters to a few centimeters), and, more noteworthily, adaptability in selecting reasonable payloads and fitting time and/or space resolutions [14]. Rendana et al. [15] recommended the use of a UAV system in small estates for the best results, given its reasonably precise coverage with low-altitude aircraft for large area coverage. By means of UAV imagery, it is possible to provide the best policies and strategies for agriculture towards desired scenarios with timely and consistent information to support stakeholder's decisions because of their flexibility in flight control, accuracy in data and signal processing, off-board sensors, and lower cost than other existing tools [16–21].

Approaches utilizing UAV imagery for tree health monitoring has gained popularity in recent years. There were several investigations in agricultural areas and test fields that were performed on forestry inventory [19,22,23] relative to fruit orchards [24–29]. Most of the imagery research utilized color and multispectral images, although the latter showed higher calibration ability and more accurate measurements for disease detection [30]. The use of UAVs has avoided many limitations related to satellite data, such as low spatial resolution, cloud spots, and long waiting times in relatively small geographical areas; nevertheless, its uses for large-scale application can be limited because it is time consuming and costly [31].

In disease-related studies employing unmanned or manned aerial systems, Calderon et al. [25] investigated the capability of thermal and hyperspectral imagery obtained by using a manned aircraft to detect early stage verticillium wilt infection in olive caused by Verticillium dahliae Kleb. The acquired images were analysed using Linear Discriminant Analysis (LDA) and Support Vector Machine (SVM) classifiers. The severity classes were divided into five categories: asymptomatic, initial, low, moderate, and severe symptoms. Their results depicted that SVM obtained higher overall accuracy (79.2%) compared to LDA (59.0%); however, the latter was better in discriminating trees at their initial stage of disease. López-López et al. [26] demonstrated the possibility of early detecting red leaf blotch (Prunus amigdalus), which is a fungal foliar disease of almond using high-resolution hyperspectral and thermal imagery, also retrieved by an aircraft. In order to discriminate between different levels of Prunus Amygdalus Dulcis severity classes (asymptomatic, initial, moderate, and high-severity), linear and nonlinear classification methods based on the forward stepwise discriminant were integrated with vegetation indices. Based on their results, linear and nonlinear models were efficient in separating healthy and severely infected trees and could be used to discriminate healthy plants from those at early disease stages. The model correctly classified 59.6% of the total sampled trees, whereas the individual classification accuracy was 64.3%, 20.0%, 70.6%, and 72.7% for asymptomatic, initial, moderate, and high-severity categories, respectively.

The spatial and spectral requirements for rapid and accurate detection of the lethal Laurel wilt disease that infected Avocado (Persea americana) were investigated by De Castro et al. [27]. They assessed a Multiple Camera Array (MCA)-6 Tetracam camera with applied filters (580, 650, 740, 750, 760, and 850 nm) from a helicopter at three different altitudes (180, 250, and 300 m) in an avocado field with two classes (healthy and infected) and four classes (healthy, early, intermediate, and late severity) systems. They also tested more than 20 vegetation indices that were potentially used to differentiate between these infection classes. For the aforementioned research purpose, they reported that the ideal flight altitude was 250 m, which resulted in 15.3 cm pixel sizes, and the optimum vegetation index was the Transformed Chlorophyll Absorption Reflectance Index 760–650 (TCARI 760–650). A research study by Smigaj et al. [28] showed the merit of using an affordable fixed-wing UAV with a thermal sensor for monitoring canopy temperature induced by needle blight infection in a diseased Scot pine. Their results indicated that the UAV-borne camera could detect the disease through temperature differences with acceptable accuracy ($R = 0.527$, $p = 0.001$). These studies have demonstrated that canopy physiological indices, such as Normalized Difference Vegetation Index (NDVI) [32], chlorophyll fluorescence, and temperature, were connected with physiological stress brought about by diseases.

García-Ruiz et al. [29] compared the use of a hyperspectral sensor AISA EAGLE VNIR Sensor (Specim Ltd., Oulu, Finland) onboard a manned aircraft, with a multiband imaging UAV-based sensor, miniMCA6 (Tetracam, Inc., Chatsworth, CA, USA). The former has 397 to 998 nm spectral range and 128 spectral bands, while the latter was a six narrow-band multispectral camera with image resolution of $1280 \times 1024$ pixels. Their study on citrus Huanglongbing diseases indicated the applicability of visible-near infrared spectroscopy at 710 nm and Near-Infrared (NIR)-R index values to discriminate between healthy or infected citrus plants, along with SVM with kernels. The accuracy of classification from the UAV based image was in the range of 67% to 85%, while for aircraft-based data, the corresponding values were slightly lower, which were from 61% to 74%.

In BSR disease-related research, the results derived from hyperspectral reflectance (non-imagery) indicated that disease infection caused by Ganoderma boninese resulted in significant reflectance changes, especially within the near infrared spectra [30,33–36], where the reflectance changes associated with this disease might as well be detected with a digital camera equipped with a NIR filter. To date, there is no report yet on the early detection of Ganoderma-infected palms from UAV platforms.

Moreover, Artificial Neural Network (ANN) has a remarkable capability to estimate complex nonlinear functions with an unknown model precisely, owing to its high effectiveness to characterize parallel distributed processing, non-linear mapping, and adaptive learning. In agricultural applications, ANN has been used to predict crop yield [37–39] among others to model fertilizer requirement [40]. In plant diseases recognition and prediction, ANNs have exhibited powerful discriminant capabilities given the sets of the best trainer [41–45]. Moshou et al. [45], for example, classified spectrograph-acquired spectral images to discriminate between healthy and yellow rusted wheat plants using quadratic discriminant analysis and ANN based algorithms. The classification accuracies of the latter were found to be better than the former, despite the fact that the overall accuracies of both classifiers were more than 90%. Wang et al. [43] compared ANN and Partial Least-Squares (PLS) models for classifying visible-IR reflectance data to identify healthy and multiple fungal-damaged soybean seeds such as downy mildew and soybean mosaic virus. The authors reported that while the PLS model could differentiate only between healthy and damaged seeds with 99% accuracy, ANN could discriminate between various fungal damages with accuracies between 84% to 100%. Sannakki et al. [46] used ANN models to classify and distinguish between downy mildew and powdery mildew on grape leaves. Through a series of image-processing techniques such as image segmentation, feature extraction, and back propagation ANN classification performed on Red–Green–Blue (RGB) images, they achieved 100% training accuracy. Zhang et al. [47] compared spectral based models developed using statistical and ANN approaches for predicting rice neck blasts

severity levels. They reported that the latter provided better accuracies of disease index in differentiating disease severity levels. Laurindo et al. [48] investigated the efficiency of several ANN algorithms for the early detection of tomato blight disease by utilizing an area under the disease progress curve as the disease indicator. Satisfactory results were produced by using the ANN given that most of the tested networks resulted in correlation values more than 0.90 between actual and predicted values.

Many scientists conducted research to detect BSR [30–35]; however, the lack of an algorithm and principles for conducting the detection of this disease within plantation areas remains a major setback. To overcome the aforementioned limitations, this study seeks to evaluate operational methods to facilitate extraction of Ganoderma-infected palms at their early stage by using a combination of UAV imagery and ANN techniques. This work is an extension of the previously published article by Ahmadi et al. [49]. The previous article applied the ANN analysis technique (Multilayer and Back-Propagation) for discriminating fungal infections at leaf scale and frond scale using spectroradiometer reflectance in oil palm trees at an early stage. However, the current work explores the potential of canopy level spectral measurements acquired from UAV imagery with the help of ANN (Levenberg–Marquardt). Moreover, the current study reveals the automatic extraction of tree crowns to calculate the mean reflectance of each individual palm. The rest of this article is organized as follows. In Section 2, the study region, datasets, and methodology are described. The results are presented in Section 3. In Section 4, a discussion is made in light of other studies; finally, Section 5 concludes this article.

## 2. Materials and Methods

### 2.1. Study Region and Datasets

The study was conducted in the Machap subdistrict belonging to United Malacca Berhad located in Melaka, Malaysia (2.402°N and 102.327°E) (WGS 84 coordinate system). These areas were previously cultivated with rubber. The fields of study were 12-year old mature oil palms planted in 2002, and census data have been conducted on 2 ha area and 374 palms (Figure 1). Weather monitoring systems have been used to observe any change in weather parameters that could contribute to plant diseases, including Ganoderma disease in oil palm. Weather data analysis was gathered from a WatchDog weather station Model 2000 (Spectrum Technologies, Inc., Aurora, IL 60504, USA) located inside the experimental field throughout the study period. The weather station provided data on temperature ($\pm 0.6$ °C), rainfall ($\pm 2\%$), and relative humidity ($\pm 3\%$).

During field data collections conducted in October 2014, we identified and marked 451 surveyed palms in the aforementioned field with four levels of infection based on specific visual symptoms on the canopy and the presence of basidiocarps on the basal of palms (Table 1). In order to confirm the presence of Ganoderma-related fungus, trunk samples were acquired from palms with the absence of Ganoderma fruiting body, and the GSM test was conducted on the samples. Based on both visual symptoms and GSM test results, the samples were segregated into four classes designated as T1 (healthy), T2 (early infected), T3 (moderately infected), and T4 (severely infected). From 451 palms, 233 were identified as T1, 38 were identified as T2, 16 were identified as T3, and 12 were identified as T4. The palms were later individually geolocated in the UAV image by using a palm triangulation map assisted by a Global Navigation Satellite System (GNSS) receiver.

**Table 1.** Classification of Ganoderma severity levels based on visual symptoms and GSM test.

| Severity Level | Symptoms |
|---|---|
| T1 (healthy) | Negative GSM test |
| | Healthy leaves and normal palm canopy |
| | Positive GSM test |
| T2 (early) | Presence of mycelium in the stem bark or brittle wood |
| | Healthy leaves and normal palm canopy |
| | Positive GSM test |
| T3 (moderate) | Presence of mycelium in the stem bark, and fruiting body |
| | Less than 50% foliar symptoms |
| T4 (severe) | Presence of fruiting body at the bottom of the rotten stem |
| | More than 50% foliar symptoms |

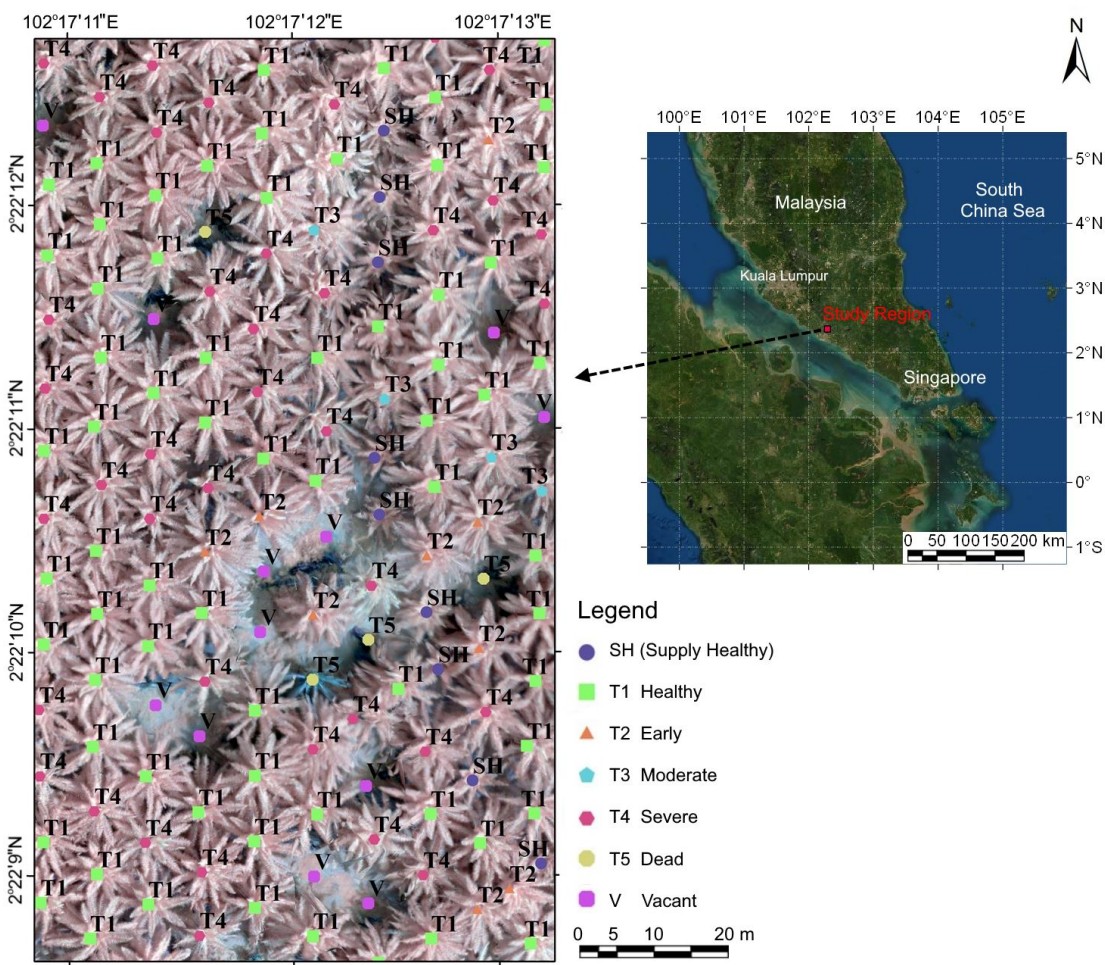

**Figure 1.** Study region. The (**right**) panel is a Google Satellite image, and the (**left**) panel shows a near-infrared image of the study region (pseudo R: NIR band; G: red band; B: green band).

## 2.2. Platform and Multispectral Camera

The employed UAV platform was a Hexacopter Tarot 680PRO folding vehicle TL68P00 that can be controlled either manually through a remote control or fully autonomously by a ground station control (Table 2). The platform was also equipped with a barometer, a magnetometer, and a Global Positioning System (GPS) antenna for calculating flight positions during the mission. The presented images in this study were obtained on 31 October 2014 by using a NIR-modified Canon Powershot SX260 HS digital camera (with external NIR filter) mounted on the aforementioned platform (Table 3). The digital camera

has three spectral bands, red, green, and NIR, and has a high-sensitivity CMOS sensor with 1–1/3200 s shutter speed.

**Table 2.** Specifications of the UAV platform.

| Type | Tarot 680PRO Folding Vehicle TL68P00 |
| --- | --- |
| Weight | 2.76 kg |
| Diameter | 695 mm |
| Battery capacity | 5200 mAh |
| Voltage | 14.8 V |
| Flying time | 14.5 min |

**Table 3.** Specifications of the digital camera.

| Camera Model | Canon PowerShot SX 260 HS |
| --- | --- |
| Resolution | $4000 \times 3000$ pixels |
| Optical zoom | 20X |
| Focal Length | 4.5 mm |
| Pixel Size | 1.5494 microns (μm) |

Prior to image acquisition, the imaging camera was calibrated and corrected using several techniques, such as Brown–Conrady model techniques [50,51] for lens distortion corrections; vignetting correction, where the image data were corrected by using per-pixel multiplication using a flat field derived correction factor called a flat field from Look-Up-Tables (LUT) factor [52–54]; and radiometric calibration based on empirical line regression models, where a tarpaulin with various colors on it was used for environmental Lambertian reflectance, also known as Pseudo-Invariant Features (PIFs) [55,56].

The Ground Sampling Distance (GSD) of imagery given a flying height of 91.3 m above ground level was 0.026 m/pixel. It covered an area of 0.089 km$^2$ that was 70% forward and 60% side overlapped. For the study region, we obtained 107 images that were later mosaicked together using Agisoft$^®$ PhotoScan (Agisoft LLC, St. Petersburg, Russia). The obtained images had Digital Numbers (DNs) in the range of 0 to 255 (8-bit). However, prior to any image manipulation and processing, the images were converted to double precision and into a double array in the range of 0 to 1 by utilizing MATLAB version 2014a.

To reduce the effects of the sensor imaging parameters, the accuracy of the camera was evaluated with the 10 Ground Control Points (GCPs) established by Real-Time Kinematic Differential Global Positioning System (RTK-DGPS) within the study region. For that reason, the GCPs polygon was overlaid on the UAV imagery, and after, that the deviation between the data was computed by using Root Mean Square Error (RMSE) to calculate accuracy, and RMSE was obtained as 0.22 m, which shows good capability for performance.

Information on surveyed palms, including their coordinates and severity levels, was overlaid on the UAV images, as shown in Figure 1, prior to image manipulation and processing. Palms marked with T1, T2, T3, and T4 represent healthy, early, moderately, and severely infected palms, respectively, whereas T5, SH, and V were dead palm, supplied healthy, and vacant palm spots, correspondingly. For classification purposes, we omitted T4 palms because these palms can be detected with human visual systems and, thus, are irrelevant to our study. We included T3 (moderately infected) palms in order to train the classification algorithms to recognize the patterns of disease severity levels.

### 2.3. Classification Using Neural Network

Our classification strategy proposed the use of ANN algorithms. Among the benefits of ANN as a classifier is the ability to model with multiple factors and complex interactions, in addition to the better ability to produce consistent predictions than methods such as regression because the former is not subject to any statistical assumptions. This is particularly valuable for the examination of dynamic phenomena concerning plant stresses [57–59].

Additionally, according to Wang et al. [59], for very high spatial resolution images, typical classification methods may not be very useful nor accurate because traditional classification does not fully utilize the spatial dimension provided by the imagery.

The ANN algorithms employed in this study were a new feed-forward network that is the most important and broadly utilized algorithm for the supervised training of the multilayer feed-forward algorithm. The operations of the feed-forward network with tansig transfer functions in the hidden layers and a purline transfer function in the output layer were performed in MATLAB version 2014a. The ANN was trained using a Levenberg–Marquardt algorithm, which is otherwise called Damped Least-Squares (DLS) method with gradient descent terms. The Levenberg–Marquardt (LM) algorithm is an iterative technique for finding the minimum of the sum of squared errors between the network outputs and the measured data [60]. This LM algorithm has exhibited not only fast training speed and robust training capability but is considerably a better classifier than other algorithms, such as the gradient descent and Gauss–Newton [61] and could decrease calculation complexity [62]. Our ANN architecture was designed to have one input layer, two hidden layers by random neuron numbers, and one output layer by two neurons. The cross-validation procedure prior to ANN model development helped to reduce the complexity of the models and minimizes the chance of models overfitting the data. The validation of the analysis was carried out by following the tenfold cross-validation method, and the average sensitivities and accuracies were obtained.

There were two subsidiary programs in this model; the task of the first one was to draw a circle around the central pixels of trees and sampling. The first step of this task was specifying the coordinates of the tree's center pixel and their disease level 285 (T1, T2, and T3); subsequently, the model created a circle where the central pixel was used as the origin. The model was also programmed to enable finding and testing different radii of the circle. The proposed method for using circle radius is an automatic identification of individual tree crowns, and tree crowns were estimated by circle fitting. In the second program, the aims were to compress the image, eliminate outlying DN values, remove zero valued pixels, and, finally, to calculate mean ($\mu$) and standard deviation ($v$) for each radius of circle. In order to reduce heterogeneity of the reflectance value within individual palms across the image, the image was resampled every four pixels in both dimensions, resulting in a 0.104 m pixel size. Additionally, the outlying DN values were discarded by eliminating values that were out of the specified range in the threshold limit. For this purpose, the threshold limit was assumed by a higher and lower ratio of rescaled DN values, for instance, if this ratio is considered an 1/8 as in this study, an 1/8 of higher and lower data was eliminated. For digitization of disease levels and input to the ANN 297 algorithm, 1 was assigned for T1, 2 for T2, and 3 for T3. The mean ($\mu$) and standard deviation ($v$) values of rescaled DN of every sample's bands and disease levels were considered as ANN model inputs and outputs, respectively. Therefore, the ANN model had four input neurons ($\mu$ and $v$ values for 2 bands).

The models with different image configurations and network properties were tested more than 200 times to determine the best neuron numbers of hidden layer, the best radius, and the best combination of spectral bands in the model. We, thus, tested the number of neurons in the hidden layers based on the structure of ANN (from 0 to 300), the modeling sample, and the dimension of network. We also experimented on different combinations of spectral bands, which were 2 layers ($\mu$ 307 and $v$ for each band) when using 1 spectral band, 4 for 2 spectral bands, and 6 for 3 spectral bands combination versus 1 output layer. The image and ANN configurations that resulted in the best overall classification accuracy of T1, T2, and T3 palms were retested to discriminate between palms of T1 and T2 only, considering that the main objective of this study was to detect early infected palms. For this purpose, the dataset was randomly split into two sets: 60% for model training and 40% for model testing. The testing dataset was also used for early stopping during model training in order to avoid model overfitting [63]. The total classification accuracy for each disease level was calculated by comparing the model predicted and actual disease level.

### 3. Results

Since our main objective is to detect Ganoderma-infected palms early or to technically discriminate between T1 and T2 palms, we emphasize our discussion on the classification accuracies of the T1 and T2 classes only. According to the classification accuracy for 287 oil palm samples that were classified into three disease levels (T1, T2, and T3), the best classification result was generated by the green and NIR bands with a circle radius of 35 pixels, 1/8 threshold limit, and ANN network of 219 hidden neurons, with a classification error of 14.29% (highlighted in Table 4).

**Table 4.** The ANN classification results for three threshold limits and three circle radii and various band combinations. Note that G, R, and N are short for Green, Red, and Near-infrared, respectively. Moreover, the circle radius (the second column in this table) is expressed as the number of pixels. The total data tested was 287 for each combination of different variables.

| Threshold | | 1/7 | | | | 1/8 | | | | 1/9 | | | |
|---|---|---|---|---|---|---|---|---|---|---|---|---|---|
| | Radius | G,R | G,N | R,N | G,R,N | G,R | G,N | R,N | G,R,N | G,R | G,N | R,N | G,R,N |
| True classified | 25 | 205 | 241 | 192 | 211 | 244 | 240 | 245 | 238 | 232 | 198 | 220 | 188 |
| | 35 | 176 | 223 | 243 | 232 | 242 | 246 | 243 | 231 | 212 | 244 | 245 | 194 |
| | 45 | 193 | 230 | 233 | 234 | 151 | 236 | 213 | 242 | 160 | 190 | 235 | 238 |
| False classified | 25 | 82 | 46 | 95 | 76 | 43 | 47 | 42 | 49 | 55 | 89 | 67 | 99 |
| | 35 | 111 | 64 | 44 | 55 | 45 | 41 | 44 | 56 | 75 | 43 | 42 | 93 |
| | 45 | 94 | 57 | 54 | 53 | 136 | 51 | 74 | 45 | 127 | 97 | 52 | 49 |
| Class error (%) | 25 | 28.57 | 16.03 | 33.10 | 26.48 | 14.98 | 16.38 | 14.63 | 17.07 | 19.16 | 31.01 | 23.34 | 34.49 |
| | 35 | 38.68 | 22.30 | 15.33 | 19.60 | 15.68 | **14.29** | 15.33 | 19.51 | 26.13 | 14.98 | 14.63 | 32.40 |
| | 45 | 32.75 | 19.86 | 18.82 | 18.47 | 47.39 | 17.77 | 25.78 | 15.68 | 44.25 | 33.80 | 18.12 | 17.07 |

Additionally, the number of ANN hidden neurons that would result in the minimum training, testing, and validation errors was simultaneously tested with the combination of the best image configurations, as previously discussed. For the sake of simplicity, we illustrated training, testing, and validation errors for a number of hidden neurons at 5, 10, 20, 30, 40, 50, 60, 70, 80, 90, 100, 110, 120, 130, 140, 150, 160, 170, 180, 190, 200, 210, 219, 220, 230, 239, 240, 250, 260, 270, 280, 290, and 300 for the combination of 1/8 threshold limit, 35 pixels circle radius, and green and NIR bands (Figure 2). The neuron numbers of 219 depicted the lowest errors of training, validation, and testing were 4.05%, 33.33%, and 26.32%, respectively (see the red-colored markers).

Further investigations on the effects of different circle radii on the mean and standard deviation of pixel values at those selected bands were performed, as shown in Figure 3. As mentioned above, all DNs were normalized and rescaled to double precision. In general, the image processing toolbox functions in the software utilized prefer DNs in the range of 0 to 1, hence explaining the magnitude of the mean values that were less than 1. There are two prominent observations that could be inferred from the table: (i) mean DN values in all bands decreased with increasing circle radius, while the standard deviations among various severity levels are approximately the same, and (ii) the maximum separation values among severity classes were noticeable at a larger circle radius. For the circle radius of 25 pixels, the mean DN value for T1 and T2 palms in all bands illustrated minimal differences, which were 0.658 and 0.661 in the green band, 0.655 and 0.657 in the red band, and 0.756 and 0.753 in the NIR band, for instance (Figure 3i). A similar observation was made for the circle radius that equaled 45 pixels, especially in the NIR band where the mean DN for T1 and T2 was 0.643 and 0.642, respectively. However, in green and red bands, the differences were slightly larger, which were 0.573 and 0.596 and 0.609 and 0.622, respectively (Figure 3iii). These values provided an inadequate separability between severity classes, especially for T1 and T2, which allows for an early detection strategy. Nevertheless, the largest separability between the mean value of T1 and T2 was observed in the circle radius of 35 pixels, for

instance, 0.613 and 0.628 in the green band, 0.608 and 0.621 in the red band, and 0.693 and 0.688 in the NIR band (Figure 3ii). It was also noticeable that standard deviation increased by increasing pixel size, despite the values between T1 and T2 being almost similar, such as 0.014 and 0.015 in the green band, for the circle radius of 25 pixels and 0.018 and 0.018 in the green band, for the circle radius of 35 pixels.

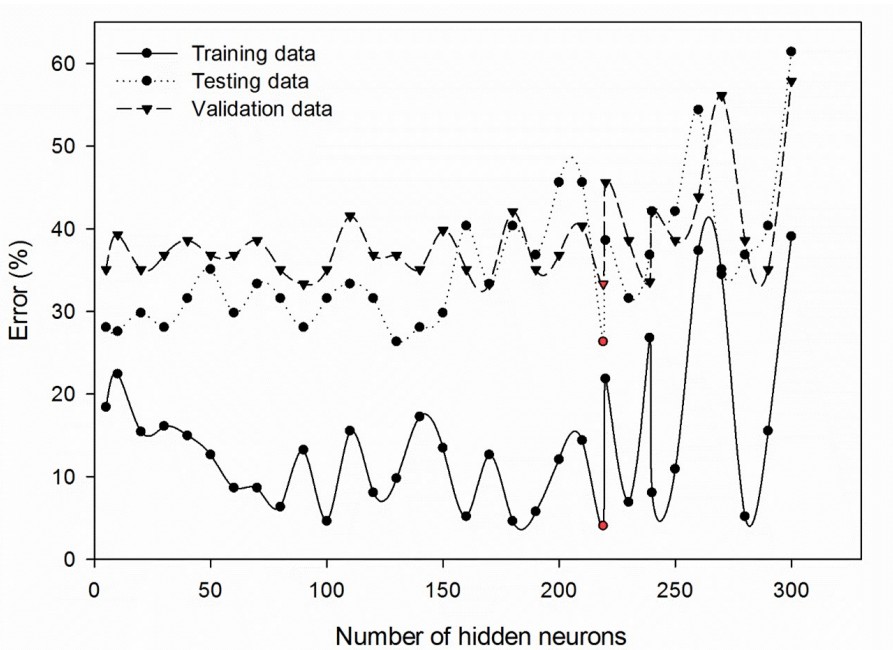

**Figure 2.** Training, testing, and validation errors for the image configurations of 1/8 threshold limit, 35 pixels circle radius, and green and NIR bands for various numbers of hidden neurons.

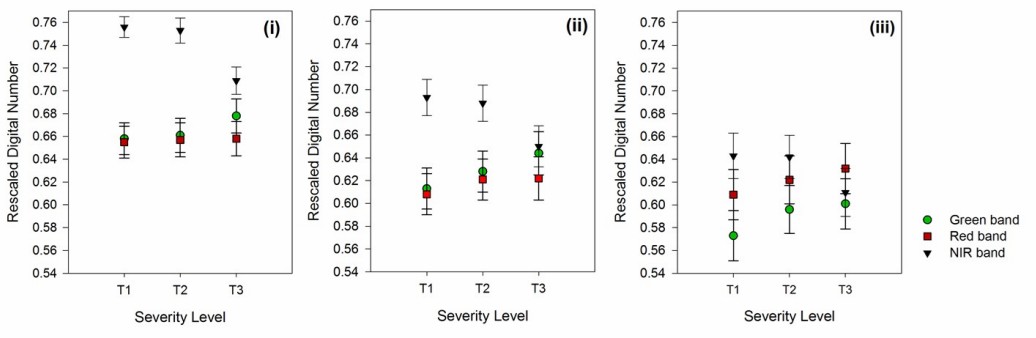

**Figure 3.** The mean and standard deviation of rescaled DN for circle radius of (**i**) 25, (**ii**) 35, and (**iii**) 45 pixels in the three bands, tested for the 1/8 threshold limit.

Investigation on threshold limit effects on the mean and standard deviation of pixel values at selected bands (Figure 4) illustrated that although increasing the threshold limit resulted in larger mean and smaller standard deviation values, such differences were negligible among the threshold limits. For instance, for the green band, the mean DN values for T1 for the threshold limit of 1/7, 1/8, and 1/9 were 0.614, 0.613, and 0.612, respectively (Figure 4i–iii). Likewise, for T2, the mean DN was 0.629, 0.628, and 0.627 for the threshold limit of 1/7, 1/8, and 1/9, respectively. It is also shown in Table 4 that the adjustment of the threshold limit did not result in better classification results.

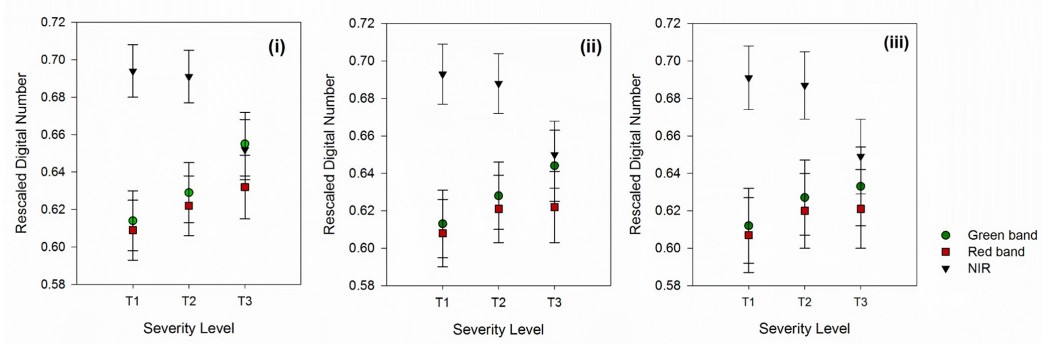

**Figure 4.** The mean and standard deviation of rescaled DN for (**i**) 1/7, (**ii**) 1/8, and (**iii**) 1/9 threshold limits in the three bands tested for the circle radius of 35 pixels.

For training and testing samples, total classification accuracies of 97.52% and 72.73% were obtained, respectively. The training model with the explained properties could detect healthy or T1 palms with 99.27% accuracy and 88.50% for T2 palms (Table 5). Nevertheless, the testing result was lower for the corresponding classes, which were 75.00% for T1 and 57.14% for T2 (Table 6).

**Table 5.** The producer accuracy for ANN training model for T1 and T2 severity level. The overall accuracy is 97.52%, and Kappa is 0.90.

| Severity Level | Total Number | True Prediction | Percentage of True Prediction |
|---|---|---|---|
| T1 | 137 | 136 | 99.27 |
| T2 | 24 | 21 | 88.50 |
| Truth overall | 161 | 157 | 97.52 |

**Table 6.** The producer accuracy for ANN testing model for T1 and T2 severity level. The overall accuracy is 72.73%, and Kappa is 0.20.

| Severity Level | Total Number | True Prediction | Percentage of True Prediction |
|---|---|---|---|
| T1 | 96 | 72 | 75.00 |
| T2 | 14 | 8 | 57.14 |
| Truth overall | 110 | 80 | 72.73 |

## 4. Discussion

The findings depicted that, for image configurations, the best combination of spectral bands to identify early infected palms was green and NIR. These results are in accordance with studies indicating that diseased or stressed vegetation showed a reflectance increment in the visible band and a decrease in NIR reflectance [64,65]. We hypothesize that the green band was more beneficial than the red one in detecting early Ganoderma-infected palms because the latter is confined to moderate to high chlorophyll content [65] and becomes easily saturated at intermediate values of leaf area index [66]. Considering that oil palm is a perennial crop and, therefore, can have a wide range of chlorophyll contents and leaf areas and the fact that BSR disease will lead to chlorosis [67] through reductions in the palms' nutrient and water uptake, the use of a green band resulted in better discrimination accuracies because it is not as easily saturated as the red band [6].

The observed patterns between the mean and standard deviations values with the circle radius and threshold limit (Figures 3 and 4) suggested that the determination of the threshold limit might not be as essential as for optimizing the circle's radius. The best classification results indicated that the circle radius has to be 140 times GSD of the original UAV image or 35 times the pixel size of the resampled UAV image, which is approximately 3.6 m from the center of the canopy, where oil palm is commonly cultivated in 9 m planting distance. Physiologically, these pixels signified that the more developed fronds could be the best approach to the early detection of BSR disease. In an oil palm tree, a newly developed

frond will emerge from the most top center of the canopy. The fronds will grow in a spiral form, either in a clockwise or counter-clockwise direction. Over time, more developed fronds will ascend to the lower part of the canopy. The younger fronds usually have a vertical angle, while more developed fronds lie horizontally. With the distance of 3.6 m from the center of canopy, signals related to Ganoderma infection are more representative from the more developed fronds that form the first spiral layer of the canopy than the young ones.

The intergroup classification accuracy of healthy palms (T1) in comparison to early infected ones (T2) was considerably better since healthier palms have more distinctive reflectance responses affected by chlorophyll quantity and structure of leaf cells that were depicted through low visible and high NIR reflectance. Therefore, with an increasing severity level of BSR disease, the accuracy of the model and the possibility of prediction by the UAV image became lower. At first glance, the validation accuracy for T2 suggested a mediocre classification capability. However, half of the T2 palms were misclassified as T1 possibly because, at an early stage of infection, a disease has not had much effect on physiological properties of plants such as on chlorophyll content [67]. Nonetheless, even with limited palm properties such as affected chlorophyll content and the absence of any visual symptoms, the ANN model and the modified NIR camera could still detect more than half of infected palms correctly.

We also compared our results to those obtained by Kresnawaty et al. [33], Liaghat et al. [35], and Lelong et al. [30], who reported 100%, 89%, and 92% accuracies for early detection of Ganoderma-infected oil palms. The latter obtained leaf spectra from a non-imaging spectroradiometer and the former utilized canopy reflectance that was also acquired from a non-imaging spectroradiometer. Better accuracies derived from these studies resulted from ground, destructive, individual leaf sampling that was time consuming and laborious compared to our rapid, non-destructive approach, yet it gained reasonable accuracies. While the use of UAV image in this study cannot justify the detection of BSR as good as ground hyperspectral equipment, this method is appropriate for field applications involving mass screening of oil palm plantations that are commonly cultivated in thousands of hectares in seeking potentially infected individual palms. Furthermore, based on another study conducted by the authors, the Support vector machine (SVM) algorithm, which is the best alternative method for non-parametric approaches and large data sets, has not achieved high accuracy for UAV image classification.

As stated earlier, previous studies were able to present high discrimination accuracy between healthy and moderately infected samples compared to the results by Shafri et al. [34] and Liaghat et al. [35]. However, the focus of this research is in mild or early stage infection where there is no visible symptom of infection that has yet appeared. To our best knowledge, no extensive study has been made to test the spectral indices on airborne hyperspectral image for the early detection of Ganoderma.

These results account for the results of previous studies that did not provide good performance in the classification of early-stage infection. Satellite multispectral remote sensing [68] had the adequate spatial resolution (2.4 m) for the crown's extraction However, its spectral resolution (60–140 nm) was inadequate for the detailed analysis of the red-edge region to allow for early detection. The aircraft-based hyperspectral remote sensing [69,70] had an adequately high spectral resolution (<10 nm) for the investigation of spectral features in the red-edge region. However, the spatial resolution (1 m) was too high to conduct pixel-based classification. If object-based classification was performed using the central portion of tree crowns, the aircraft-based hyperspectral remote sensing could have a better result for early detection. Lastly, the previous research method by Ahmadi et al. [49] was time consuming and destructive compared to the current study, which is rapid, non-destructive, and on the canopy level.

## 5. Conclusions

This study suggests that remote sensing images acquired by using an affordable, modified digital camera mounted on a UAV platform, combined with the ANN algorithms, have great potential for oil palm disease detection under field conditions. Our results showed that this type of platform can be used effectively in an early detection a BSR disease program. The nature of the UAV images that have a very high spatial resolution despite being highly heterogeneous was able to accommodate the need for detection of Ganoderma infection at its earliest stages by providing information related to the most Ganoderma-affected fronds. Moreover, the employment of ANN classification algorithms was successful in modeling the subtle differences between the rescaled DNs of healthy and early infected palms.

In this work, an ANN-based classifier, specifically a feed-forward ANN network with tansig transfer functions in the hidden layers and a purline transfer function in the output layer trained using a Levenberg–Marquardt algorithm, was employed in achieving its objective to recognize and classify early Ganoderma-infected palms. The principle of the classification is to seek the most representative mean and standard deviation values from green, red, and NIR bands, while adjusting for the best circle radius and threshold limit. Simultaneously, a number of hidden neurons and termination error optimization were optimized by providing various classification inputs in order to correctly classify the imaged palms to their corresponding severity classes. The best protocol in classifying pre-symptomatic palms infected with BSR disease is with an ANN network by 219 hidden neurons and 0.00001 termination error rate, green and NIR bands, circle radius of 35 pixels, and 1/8 threshold limit, with respect to the image normalization and compression step. In summary, this study demonstrated the use of UAV-based hyperspectral imaging and the ANN approach for the early detection of BSR disease infection in oil palm trees. Random Forest and/or other AI techniques can also be useful and are recommended in the next study.

**Author Contributions:** Writing—original draft preparation, P.A.; writing—review and editing, S.M., B.F. and E.G. All authors have read and agreed to the published version of the manuscript.

**Funding:** This research received no external funding.

**Institutional Review Board Statement:** Not applicable.

**Informed Consent Statement:** Not applicable.

**Data Availability Statement:** The datasets used in this study are available from the corresponding author on reasonable request.

**Acknowledgments:** We would like to thank the United Malacca Berhad for the financial support and providing the study region to conduct this study. We also thank the editors and reviewers for their time and constructive comments that helped us to improve the presentation of the paper. All Malaysian estates and mills are certified with the Malaysian Sustainable Palm Oil (MSPO) certification (accessed on 28 February 2022): https://unitedmalacca.com.my/sustainability/overview/.

**Conflicts of Interest:** The authors declare no conflict of interest.

## Abbreviations

The following abbreviations are used in this manuscript:

| | |
|---|---|
| ANN | Artificial Neural Network; |
| BSR | Basal Stem Rot; |
| DLS | Damped Least-Squares; |
| DN | Digital Number; |
| EDTA | Ethylenediamine Tetra-acetic Acid; |
| GCP | Ground Control Point; |

| | |
|---|---|
| GNSS | Global Navigation Satellite System; |
| GPS | Global Positioning System; |
| GSD | Ground Sampling Distance; |
| GSM | Ganoderma Selective Media; |
| LDA | Linear Discriminant Analysis; |
| LM | Levenberg–Marquardt; |
| LUT | Look-Up-Tables; |
| MCA | Multiple Camera Array; |
| NDVI | Normalized Difference Vegetation Index; |
| PCR | Polymerase Chain Response; |
| PIF | Pseudo-Invariant Feature; |
| PLS | Partial Least-Squares; |
| RMSE | Root Mean Square Error; |
| RTK-DGPS | Real-Time Kinematic Differential GPS; |
| SVW | Support Vector Machine; |
| TCARI | Transformed Chlorophyll Absorption Reflectance Index; |
| UAV | Unmanned Aerial Vehicle; |
| VOC | Volatile Organic Compound. |

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
