# Peer review of "Unmanned Aerial Vehicle (UAV)-Based Remote Sensing for Early-Stage Detection of Ganoderma"

_remotesensing, doi:10.3390/rs14051239_

Round 1

Reviewer 1 Report

The work is interesting, a review of the literature, methodology, description of results is almost perfect and correct, but with minor errors. The manuscript is well prepared. However, there is an important concern that has already arisen in previous reviews and has not been explained fully. The study site, methodology and input data were taken from another publication by the author from 2017 year. The authors do not mention this in the methodology section.  It can therefore be considered that self-plagiarism may have occurred. The authors only mention their earlier work in the discussion chapter. There are some differences in the analysis of the images between the papers, but it should be the responsibility of the authors to demonstrate these differences to avoid accusations of self-plagiarism, not the reviewers. The authors should demonstrate and make clear what the differences between the papers actually are and what the added value is.

Minor shortcomings:

Line 75: „red leaf blotch” Incorrect Latin name. Prunus Amygdalus Dulcis is the name of the carrier tree, not the pathogen

Author Response

Dear Reviewer,

We thank you very much for your time and constructive comments. We have added a paragraph in the Introduction to describe how this paper is different from the earlier work. The changes are highlighted in the revised version. Please also see the attached file.

We hope that the changes that we made are satisfactory and we appreciate your comments.

Sincerely yours,
Authors

Reviewer 2 Report

Comments to the author:

According to adjust for the most representative mean and standard deviation values from green, red, and near-infrared bands, as well as best palm circle radius, threshold limit and number of hidden neurons for different Ganoderma severity levels, this manuscript has constructed a rapid detection of Ganoderma infected oil palms Unmanned Aerial Vehicle (UAV) imagery integrated with an Artificial Neural Network (ANN) model. Better accuracies derived from these studies resulted from ground, destructive, individual leaf sampling that was time-consuming and laborious, compared to the classification which are rapid, non-destructive approach, yet gained at reasonable accuracies. The article is well-focused and the description is completely narrated, but there are still some problems remaining to be explained, which is as followed:

Q1: The classification strategy proposed the use of ANN algorithms. Despite the benefits of ANN as a classifier is the ability to model with multiple factors and complex interactions, plus the better ability to produce consistent predictions. Whereas, random forest algorithm may also achieve good accuracy. In the training process, the interaction between features can be detected and it can give which features are important after the training. So we can compare ANN algorithm with random forest algorithm, which will make the article abundantly.

Q2: There were two subsidiary program in this model, the first step of this model was also programmed to enable finding and testing different radii of the circle. However, why the model does this selection and manipulation of the radius of the circle is not specified. How does the radius of the circle affect, it should be clear.

Author Response

Dear Reviewer,

We thank you very much for your time and constructive comments. The changes are highlighted in the revised version. Please also see the attached file.

We hope that the changes that we made are satisfactory, and we appreciate your comments.

Sincerely yours,
Authors

Round 2

Reviewer 1 Report

Thank you for responding to my review. Based on my own comparative analysis of the two documents: i.e. the 2017 publication and the one presented now, I do not find that self-plagiarism has occurred.

Unfortunately, the additional description proposed by the authors does not give a precise description of the differences, all the differences, and the explanation is not clear. The authors need to emphasise even more strongly the differences between the two works. This is important! They are there, but the reader must not guess at them, it should be given to him directly.

Other parts of the manuscript meet the requirements for publication.

Details:

“This work is an extension of the previously published article by Ahmadi et al. [49].

This sentence is clear, without qualification.

“The previous article applied the ANN analysis technique for discriminating fungal infections using spectroradiometer reflectance in oil palm trees at an early stage as well as providing new insight on the best frond number and the applicability of data collection under different weather conditions while the current work explores potential of canopy level spectral measurements acquired from UAV imagery with the help of ANN. “

The sentence is too long, make it two. Plus we don't know if the "new insights" are in the 2017 paper or this one.

The 2017 paper and the current paper use different ANN algorithms. 2017 - "multilayer and back-propagation", now - "Levenberg-Marquardt" . Probably worth mentioning.

Author Response

Dear Reviewer,

We appreciate your time and constructive comments. We have modified the paragraph in the Introduction based on your comments. The changes are

highlighted in the revised version.

In lines 152-158 of the revised version, we wrote:

“This work is an extension of the previously published article by Ahmadi et al. [49]. The previous article applied the ANN analysis technique (Multilayer and Back-Propagation) for discriminating fungal infections at leaf scale and frond scale using spectroradiometer reflectance in oil palm trees at an early stage. However, the current work explores the potential of canopy level spectral measurements acquired from UAV imagery with the help of ANN (Levenberg-Marquardt). Moreover, the current study reveals automatic extraction of tree crowns to calculate the mean reflectance of each individual palm.”

We hope that the changes that we made are satisfactory and we appreciate your comments.

Sincerely yours,

Authors
